# Quantum interference directed chiral raman scattering in two-dimensional enantiomers

Shishu Zhang[1,4], Jianqi Huang[2,3,4], Yue Yu[1], Shanshan Wang[1], Teng Yang [2,3✉], Zhidong Zhang [2,3], Lianming Tong [1✉] & Jin Zhang [1]

Raman scattering spectroscopy has been a necessary and accurate tool not only for characterizing lattice structure, but also for probing electron-photon and electron-phonon interactions. In the quantum picture, electrons at ground states can be excited to intermediate energy levels by photons at different **k**-points in the Brillouin zone, then couple to phonons and emit photons with changed energies. The elementary Raman processes via all possible pathways can interfere with each other, giving rise to intriguing scattering effects. Here we report that quantum interference can lead to significant chiral Raman response in monolayer transitional metal dichalcogenide with triclinic symmetry. Large circular intensity difference observed for monolayer rhenium dichalcogenide originates from inter-**k** interference of Raman scattering excited by circularly polarized light with opposite helicities. Our results reveal chiral Raman spectra as a new manifestation of quantum interference in Raman scattering process, and may inspire induction of chiral optical response in other materials.

[1] Center for Nanochemistry, Beijing Science and Engineering Center for Nanocarbons, Beijing National Laboratory for Molecular Sciences, College of Chemistry and Molecular Engineering, Peking University, Beijing, China. [2] Shenyang National Laboratory for Materials Science, Institute of Metal Research, Chinese Academy of Sciences, Shenyang, China. [3] School of Material Science and Engineering, University of Science and Technology of China, Shenyang, China. [4]These authors contributed equally: Shishu Zhang, Jianqi Huang. ✉email: yangteng@imr.ac.cn; tonglm@pku.edu.cn

D uring a Raman scattering process, phonons are coupled to photo-excited electrons that relax to the ground state by the emission of scattered photons[1–3]. The frequency of a Raman mode is given by the energy difference between the incident and scattered photons[4], and reveals structural information of materials[5,6]. On the other hand, the intensity is more sophisticated to estimate, and is majorly determined by the electron–photon and electron–phonon interactions[7,8]. More importantly, quantum interference can occur between different elementary pathways of Raman scattering, resulting in the modulation of inelastic scattering efficiencies[9–11]. So far, the quantum interference effect in Raman scattering have only been reported for few two-dimensional (2D) layered materials, such as electrostatically doped graphene[12] and few-layer $MoTe_2$[13,14]. The observation of the quantum interference is challenging, yet essential for understanding the fundamental light-matter interaction, and for possible manipulation of light scattering in materials.

Rhenium dichalcogenide ($ReX_2$, X = S or Se) is a layered transitional metal dichalcogenide (TMDC) material with triclinic symmetry[15]. The Re atoms in $ReX_2$ (X = S, Se) crystals move away from metal sites in 2H phase to form $Re_4$ parallelograms, leading to a distorted crystal structure and giving rise to anisotropic in-plane properties, such as anisotropic carrier mobility[16–18], photoluminescence[7,19,20] and Raman scattering[21,22]. It has been reported that chiral Raman scattering arose in $ReS_2$ flakes of tens of nanometers in thickness, where the anisotropic optical effects play a prominent role[23]. However, the underlying fundamental interactions between photons/electrons/phonons during the chiral Raman scattering process, apart from the optical effects, have remained unexplored.

Herein, we report that chiral Raman scattering can be observed in monolayer $ReS_2$ and $ReSe_2$ excited by circularly polarized light, and large circular intensity difference (CID) is induced by the quantum interference between first-order Raman processes occurred at different **k**-points in the Brillouin zone. Our calculations show that although the amplitudes of induced electric dipoles are the same at each **k**-point, a phase difference exists for left- and right-handed circular polarization excitations, which results in the circular polarization differentiated Raman scattered intensities for all Raman modes due to quantum interference effect. The calculated chiral Raman spectra based on the quantum interference effect agree well with the experimental ones.

## Results
**Circular intensity difference in 2D $ReS_2$ enantiomers**. The optical setup for chiral Raman scattering measurements is shown in Fig. 1a. A quarter wave plate (QWP) is used to produce right-handed (RCP) or left-handed (LCP) circular polarization for excitation. The scattered light passes through the same QWP, and is collected without any analyzer. Single-layer (1L) $ReS_2$ was mechanically exfoliated on a fused silica substrate (170 μm in thickness). The optical and atomic force microscopy (AFM) images are shown in Supplementary Fig. 2. The thickness of monolayer $ReS_2$ was measured to be 1.1 nm.

The eigenvectors of the Raman-active vibrational modes (marked as mode I–VI) were calculated using density functional theory (DFT) and are shown in Fig. 1b[24,25]. In Fig. 1c, d, we show the experimental Raman spectra of 1L $ReS_2$ below 250 cm$^{-1}$, excited by RCP and LCP of 1.96 eV and 2.33 eV, respectively. The helicity-dependent Raman intensities of 1L $ReS_2$ can be clearly seen, and the intensity differences ($I_R - I_L$) are depicted in the insets. When excited by 1.96 eV photon energy, both mode III (153 cm$^{-1}$) and mode VI (235 cm$^{-1}$) exhibit obvious chiral response but with opposite signs. For 2.33 eV excitation photon

energy, the Raman scattering efficiencies by RCP are higher than that by LCP for I (132 cm$^{-1}$), II (143 cm$^{-1}$), III (153 cm$^{-1}$), and V (212 cm$^{-1}$) Raman modes, while mode IV at 162 cm$^{-1}$ and mode VI at 235 cm$^{-1}$ show opposite chiral Raman response. The circular intensity differential (CID, △) values, defined as ($I_R - I_L$)/($I_R + I_L$), are obtained to be 0.49, 0.33, 0.18, −0.05, 0.27 and −0.16 for modes I–VI, respectively, for 2.33 eV excitation. While with 1.96 eV excitation photon energy, the CID values are −0.43 for 153 cm$^{-1}$ mode and 0.34 for 235 cm$^{-1}$ mode. The CID values are distinctly correlated to the phonon modes with certain excitation photon energy. This indicates that the helicity-dependence does not originate from circular dichroism[26]. It should also be noted that owing to the monolayer thickness (~1 nm), the optical birefringence can be negligible[27].

Owing to the triclinic symmetry, layered $ReX_2$ has two vertical orientations[23,28]. The chiral Raman scattering spectra of monolayer $ReS_2$ with two different orientations were also measured and depicted in Fig. 2. To distinguish the two orientations, the atomic structures of 1L $ReS_2$ were characterized by annular dark-field (ADF) scanning transmission electron microscopy (STEM) and the results are shown in Fig. 2a, e, where the $Re_4$ parallelograms are highlighted. The vertical orientation in Fig. 2a is defined as $ReS_2$ (+), and the angle between the *a-axis* ([100]) and *b-axis* ([010]) is 119.8° (anticlockwise), while for $ReS_2$ (−) in Fig. 2e, the angle is −119.8° (clockwise).

To further investigate the chiral Raman scattering, the scattering efficiencies of $ReS_2$ (+) and $ReS_2$ (−) were measured as a function of rotation angle of the QWP. The Jones matrix of the electric field can be described as $\begin{pmatrix} 1 - i\cos2\theta & -i\sin2\theta \\ -i\sin2\theta & 1 + i\cos2\theta \end{pmatrix}$. The Raman scattering intensities excited by 1.96 eV (Fig. 2b) and 2.33 eV (Fig. 2c) photon energy vary periodically. The intensities of all the Raman modes with different rotation angles $\theta$ are extracted and the polar plots are depicted in Supplementary Tables 3, 4. Figure 2d shows the polar plots of mode 132 cm$^{-1}$. For 1.96 eV excitation, the Raman scattering intensity of RCP ($I_R$) is almost identical to that of LCP ($I_L$). While for 2.33 eV excitation, the 132 cm$^{-1}$ Raman peak shows the maximum $I_R$ and the minimum $I_L$. These results imply that the chiral Raman response is dependent on the excitation photon energy. For $ReS_2$ (−), the intensity maps (Fig. 2f, g) and the polar plots of the 132 cm$^{-1}$ mode in Fig. 2h show opposite trends to that of 1L $ReS_2$ (+).

It has been reported that chirality in two-dimensions arises when distinct enantiomers are confined to a plane, that can be interconverted by space inversion but owing to the inaccessible third dimension, can not be interconverted by any rotation about the axis perpendicular to the plane[29,30]. Considering the spatial operation in three dimensions, $ReS_2$ is an achiral material. However, layered $ReS_2$ confined to a substrate exhibits planer chirality, owing to that the mirror reflection of $Re_4$ parallelogram can not be superposed on its original one by rotation (Supplementary Fig. 1). Indeed, the lack of inversion center in the 2D plane leads to the two distinct 2D $ReS_2$ enantiomers. However, it is still not clear how the chiral Raman response arises from the microcosmic view of electron/photon and electron/phonon interactions.

**Theoretical calculations**. In order to understand the physical origin of large CID observed in 1L $ReS_2$, the intensity of the first-order Raman scattering is calculated by third-order perturbation theory. The Raman intensity and Raman tensor can be both calculated according to the electronic band structure and the phonon dispersion (Supplementary Fig. 3) of 1L $ReS_2$, and the expressions are given in Eqs. (1) and (2) in the Method section, respectively. The calculated band gap (~1.50 eV) is indirect in the

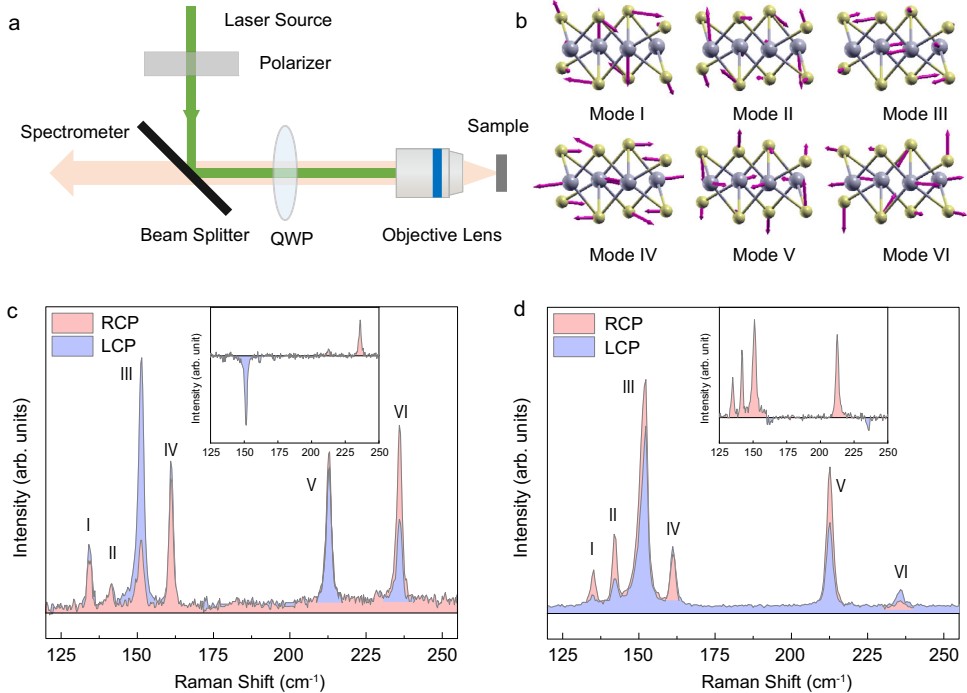

**Fig. 1 Chiral Raman response of monolayer ReS₂. a** Optical setup of the chiral Raman scattering measurements; **b** The eigenvectors of the Raman-active vibrational modes (I–VI) according to density functional theory (DFT) calculations; **c, d** Circularly polarized Raman spectra for the Re vibrational modes of 1L ReS₂ ($E_L = 1.96$ eV **c** and 2.33 eV **d**). The insets depict the difference between the Raman intensities of 1L ReS₂ excited by RCP and LCP.

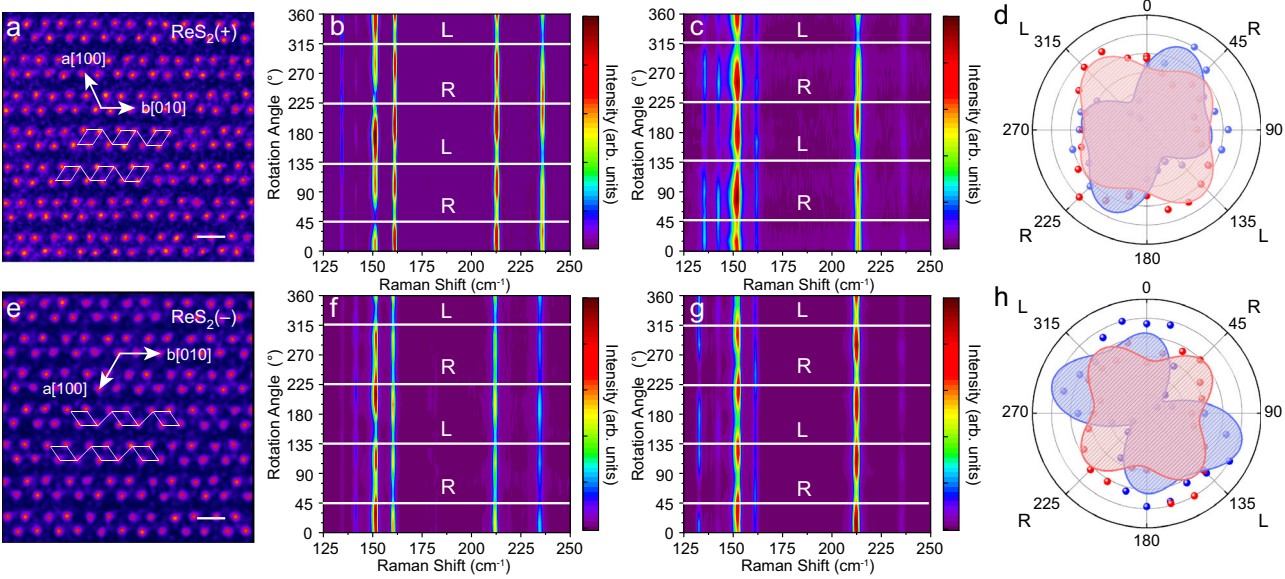

**Fig. 2 Circularly polarized Raman spectra of ReS₂ with different vertical orientations. a, e** STEM image of ReS₂ (+) and ReS₂ (−), scale bar: 0.5 nm. **b, c, f, g** Raman intensities for different rotation angles excited by 1.96 eV **b, f** and 2.33 eV **c, g** lasers. The polarization states for excitation are indicated by white lines, where L and R refer to left-handedness and right-handedness. **d, h** Polar plots of normalized Raman intensity of 132 cm⁻¹ mode excited by 1.96 eV (red) and 2.33 eV (blue) for ReS₂ (+) **d** and ReS₂ (−) **h**.

vicinity of Brillouin zone center and the direct one is around 1.55 eV, which is almost coincident with the experimental fundamental optical absorption edge in literature[31]. The calculated phonon dispersion relation shows dispersive phonon bands below 250 cm⁻¹ and relatively flat bands above 250 cm⁻¹. The phonon modes of interest here lie below 250 cm⁻¹, as marked in yellow. There are 9 optical phonon modes in this region in which all the 6 gerade modes ($A_g$) are Raman-active while the other 3 ungerade modes ($A_u$) are infrared-active according to the character table of

$C_i$ point group. Raman frequencies of the 6 Raman modes agree quantitatively between the experiment and calculation. More importantly, the calculated Raman spectra reveals that the chiral Raman scattering of 1L ReS₂ arises from the interference effect between all possible quantum pathways of elementary transitions as discussed below.

Figure 3 shows the dependence of the calculated Raman spectra on laser energy and helicity. In Fig. 3a, b, the Raman spectra within three different interference patterns excited by both RCP

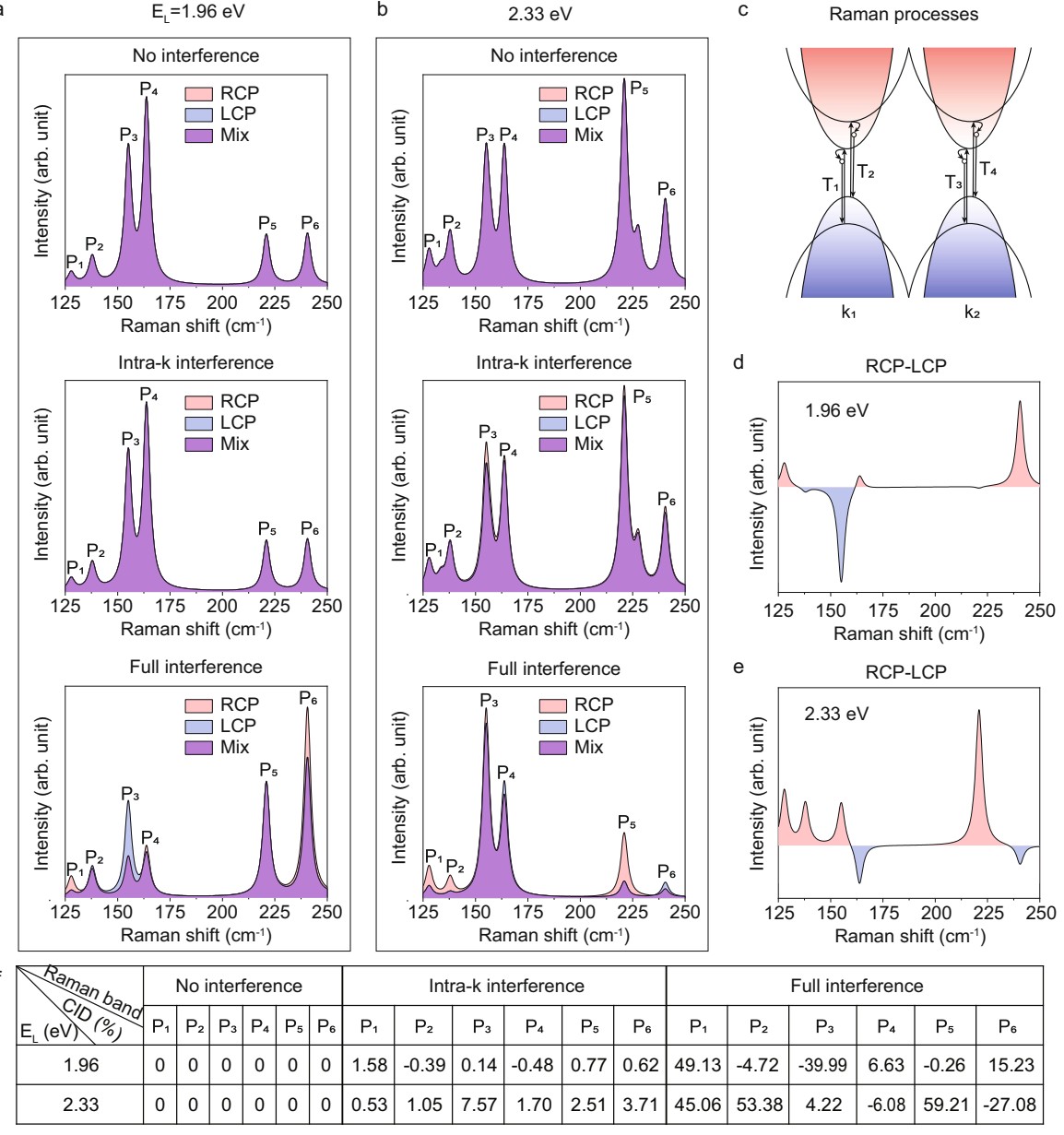

**Fig. 3 Calculated results of chiral Raman scattering in 1L ReS₂.** Raman spectra from three different interference pathways (no interference, intra-k interference, and full interference) excited by both RCP and LCP for **a** 1.96 eV **b** 2.33 eV photon energies. **c** Schematics of different quantum pathways in Raman process. **d**, **e** Calculated CID spectra for 1.96 eV and 2.33 eV photon energies. **f** The calculated CID values of the six Raman modes for the three interference patterns.

and LCP at two laser energies are given. The three interference patterns include no interference, intra-**k** interference, and full interference, as demonstrated in Fig. 3c. In the simplest scenario, 4 electrons at two **k**-points are assumed to be involved in the Raman scattering process, for each scattering channel $T_n = e_s * R_n * e_i$, in which $R_n$ is the Raman tensor, $e_i$ and $e_s$ is Jones vector of incident and scattering polarization, respectively. No interference refers to the case that Raman intensity consists of simple superposition of each independent Raman process, namely, $|T_1|^2 + |T_2|^2 + |T_3|^2 + |T_4|^2$; intra-**k** interference to $|T_1 + T_2|^2 + |T_3 + T_4|^2$, and full interference to $|T_1 + T_2 + T_3 + T_4|^2$ (consistent with the third-order perturbation theory in Eq. (1)). From top to bottom panel of Fig. 3a, b, Raman spectra excited by two circularly polarized lasers show no difference between RCP and LCP for no interference case; the intra-**k** interference shows very slight difference but CID values do not

agree with the experimental data; when the full interference effect is concerned, not only the CID sign but also the relative values agree well with experimental observations. The calculated CID spectra for 1.96 eV and 2.33 eV lasers are given in Fig. 3d, e. To have a direct impression, the calculated values of CID for the 6 Raman modes at two photon energies in the three interference patterns are listed in Fig. 3f. The agreement between the calculated CID spectra in Fig. 3d, e and our experimental measurement in Fig. 1c, d reveals the essential role played by the full interference between all possible quantum pathways of elementary transitions. The microscopic picture of the full interference effect and its relevance to the phase coherent dipole vector **D** are discussed in more details in Supplementary Material 4. Briefly, the amplitudes of induced dipole vectors are the same for LCP and RCP excitations, so that the optical absorption is helicity-independent (Supplementary Fig. 4a, b). This confirms

that no circular dichroism is expected in ReS$_2$. Further, the phase angles of Raman tensors $\mathbf{R_{i,f,n,n'}(k)}$ involving one electron are found to be either parallel or antiparallel to each other (Supplementary Fig. 5), indicating that the Raman intensity would not depend on helicity if no inter-$\mathbf{k}$ interference effect considered. However, the $\mathbf{k}$-resolved phase distribution of complex $\mathbf{T_{i,f,n,n'}(k) = e_s \cdot R_{i,f,n,n'}(k) \cdot e_i}$ is different between LCP and RCP excitations, leading to different interference between $\mathbf{k}$ points, and hence distinctive Raman intensities as observed in Fig. 1c, d and calculated in Fig. 3d, e.

The Raman tensor of ReS$_2$ for a given excitation photon energy, defined as $\begin{pmatrix} a e^{i\varphi_a} & b e^{i\varphi_b} \\ b e^{i\varphi_b} & c e^{i\varphi_c} \end{pmatrix}$ for 2D ReS$_2$ (+), is the result of the quantum interference between all $\mathbf{k}$-points. The tensor elements are calculated by considering full interference and are listed in Supplementary Tables 1, 2. The 2D ReS$_2$ (−) is the mirror reflection of ReS$_2$ (+), and the first Brillouin zone is the primitive cell of the reciprocal lattice, hence it can not be interconverted with its mirror reflection, resulting in enantio-morphous electronic band structure and phonon dispersion. Therefore, the Raman tensor for ReS$_2$ (−) is represented as $\begin{pmatrix} a e^{i\varphi_a} & -b e^{i\varphi_b} \\ -b e^{i\varphi_b} & c e^{i\varphi_c} \end{pmatrix}$. The Raman intensity of 2D ReS$_2$ (+) for LCP and RCP excitations can be evaluated by the Raman selection rules as:

$$I_L \propto \pi\left(a^2 + 2b^2 + c^2\right) - 2\pi\left(ab\sin\left(\varphi_b - \varphi_a\right) - bc\sin\left(\varphi_c - \varphi_b\right)\right),$$

$$I_R \propto \pi\left(a^2 + 2b^2 + c^2\right) + 2\pi\left(ab\sin\left(\varphi_b - \varphi_a\right) - bc\sin\left(\varphi_c - \varphi_b\right)\right).$$

However, for ReS$_2$ (−), $be^{i\varphi_b}$ is replaced by $-be^{i\varphi_b}$, leading to opposite chiral Raman responses $\Delta(+) = -\Delta(-)$.

**Chiral Raman response of ReSe$_2$.** Although the quantum interference effect in Raman scattering has been reported in other 2D materials[11–14], this is the first observation of quantum interference leading to pronounced chiral Raman response. Furthermore, it can be expected that quantum interference exists in Raman scattering of a broader frame of materials. Here, we show that the chiral Raman scattering can also be observed in other triclinic 2D layered materials such as ReSe$_2$. The crystal structure of ReSe$_2$ is similar to ReS$_2$, and there are 18 Raman active modes ($A_g$ modes) in the center of Brillouin zone[32]. The optical and AFM images of mechanically exfoliated 1L ReSe$_2$(±) samples are shown in Supplementary Fig. 2e–h. Figure 4a show the chiral Raman scattering spectra ($E_L$ = 2.33 eV) of mechanically exfoliated 1L ReSe$_2$ (+). The corresponding ADF-STEM image is given in Fig. 4b, where the Re$_4$ structure is indicated by white-parallelogram. The Raman intensities of the four modes at 110 cm$^{-1}$, 119 cm$^{-1}$, 125 cm$^{-1}$, and 162 cm$^{-1}$ are extracted for different excitation photon energies and plotted in Supplementary Tables 5, 6, respectively. The modes at 119 cm$^{-1}$ and 162 cm$^{-1}$ are shown in Fig. 4c, d with 2.33 eV in blue and 1.96 eV in red color. From Fig. 4c, it can be seen that for 2.33 eV excitation, $I_R$ of 119 cm$^{-1}$ mode is larger than $I_L$, while for 1.96 eV, $I_L$ is larger than $I_R$. These results also indicate the occurrence of quantum interference in Raman scattering. For the 162 cm$^{-1}$ mode shown in Fig. 4d, $I_R$ is larger than $I_L$ for 2.33 eV, but almost the same for 1.96 eV. The results for ReSe$_2$(−) are given in Fig. 4e–h. The 119 cm$^{-1}$ mode (Fig. 4g) shows larger $I_L$ for 2.33 eV, and larger $I_R$ for 1.96 eV, and the fitting profiles of the polar plots are opposite to that of ReSe$_2$ (+). As for the 162 cm$^{-1}$ mode, the chiral Raman response is also orientation-dependent (Fig. 4h). For 2.33 eV excitation, the 162 cm$^{-1}$ mode has the lowest $I_R$ and significantly larger $I_L$, but $I_R$ is larger than $I_L$.

## Discussions

In summary, we reported that the quantum interference in Raman scattering can result in strong chiral response in 2D enantiomers of single-layer triclinic ReX$_2$. The Raman scattering efficiencies for LCP and RCP excitations are observed to be clearly different and depend on the Raman modes and excitation photon energies. Our calculations reveal that the optical absorption intensities are exactly the same in the whole $\mathbf{k}$-space for photons of LCP and RCP, however, the $\mathbf{k}$-resolved phases of complex $\mathbf{T_{i,f,n,n'}(k) = e_s \cdot R_{i,f,n,n'}(k) \cdot e_i}$ show different patterns, resulting in distinct Raman scattering intensities as a result of different quantum interference.

Our findings reveal that quantum interference can lead to pronounced chiral response of Raman scattering in materials and indicate that quantum interference can be a generic effect in inelastic optical scattering, which becomes evident when either constructive or destructive interference between all the inelastic scattering pathways dominates, in the condition that the excitation photon energy is larger than the band gap of the materials. This effect is also applicable to the bulk triclinic crystals (ReS$_2$ and ReSe$_2$), but additional anisotropic optical environment should be taken into consideration for a complete analysis of chiral response.

## Methods

**Specimen preparation and Raman scattering measurement.** The ReS$_2$ and ReSe$_2$ flakes were mechanically exfoliated on fused silica substrates. Single-layer ReS$_2$ and ReSe$_2$ were located by an optical microscope (Olympus BX51), and the thicknesses were measured by atomic force microscope (AFM, Bruker ICON). Raman spectra were measured using JY Horiba HR800 with 2.33 eV and 1.96 eV excitation energy at room temperature. A 100× objective lens (NA0.9) and 1800 lines/mm grating were chosen for spectra acquisition.

**STEM measurement.** A PPC/PDMS stamp was used to pick the ReS$_2$ flakes from the substrates, and was covered onto the SiN$_x$ grids using a transfer stage. The temperature was raised to 110 °C until the stamps and the grids were well con-tacted, and then the grids and the stamps were placed in acetone for 24 h at room temperature to remove the PPC. STEM-ADF images were taken using FEI Titan Cubed Themis G2 300 operated at 300 kV. The convergence semi-angle was 21.3 mrad while the collection angle of ADF detector was 39–200 mrad. While acquiring images, the probe current was ~8 pA, and the dwell time was 2 μs/pixel. For this condition, the radiation damages can be avoided and images with high signal-to-noise ratio were obtained.

**Computation method.** First-order Raman intensity and Raman tensor as a function of phonon energy $\hbar\omega_\nu$ and of the incident laser energy $E_L$ can be described by third-order perturbation theory as

$$I^\nu(E_L) \propto \left| \sum_\mathbf{k} \sum_{i=f,m,m'} \frac{\mathbf{M_{opt}^{fm'e_s}(k) \cdot M_{ep}^{m'm_\nu}(k) \cdot M_{opt}^{mie_i}(k)}}{\left(E_L - \Delta E_{mi} - i\gamma\right)\left(E_L - \hbar\omega_\nu - \Delta E_{m'i} - i\gamma\right)} \right|^2 \quad (1)$$

$$R(\nu) = \sum_k \sum_{i=f,m,m'} \frac{\mathbf{D^{fm'}(k) \cdot M_{ep}^{m'm_\nu}(k) \cdot D^{mi}(k)^*}}{\left(E_L - \Delta E_{mi} - i\gamma\right)\left(E_L - \hbar\omega_\nu - \Delta E_{m'i} - i\gamma\right)} \quad (2)$$

where $\mathbf{D^{fm'}(k)}$ and $\mathbf{D^{mi}(k)^*}$ are the dipole vectors for photon emission and absorption in the electron-photon interactions, respectively.

To single out the interference effect, we use Eq. (1) for the full interferences between all possible quantum pathways. To compare, Eq. (3), is used for considering the interference only between different excited states at the same k point not between different k points, and Eq. (4) for the case of no interference at all.

$$I^\nu(E_L) \propto \sum_k \left| \sum_{i=f,m,m'} \frac{\mathbf{M_{opt}^{fm'e_s}(k) \cdot M_{ep}^{m'm_\nu}(k) \cdot M_{opt}^{mie_i}(k)}}{\left(E_L - \Delta E_{mi} - i\gamma\right)\left(E_L - \hbar\omega_\nu - \Delta E_{m'i} - i\gamma\right)} \right|^2 \quad (3)$$

$$I^\nu(E_L) \propto \sum_k \sum_{i=f,m,m'} \left| \frac{\mathbf{M_{opt}^{fm'e_s}(k) \cdot M_{ep}^{m'm_\nu}(k) \cdot M_{opt}^{mie_i}(k)}}{\left(E_L - \Delta E_{mi} - i\gamma\right)\left(E_L - \hbar\omega_\nu - \Delta E_{m'i} - i\gamma\right)} \right|^2 \quad (4)$$

All the required matrix elements are calculated by using a modified version of the QUANTUM-ESPRESSO code, and especially, the electron-phonon coupling is computed by means of the Wannier interpolation schemes in the standard Electron–Phonon Wannier (EPW) package as implemented in QUANTUM-ESPRESSO. The electron and phonon bands are calculated in the local density

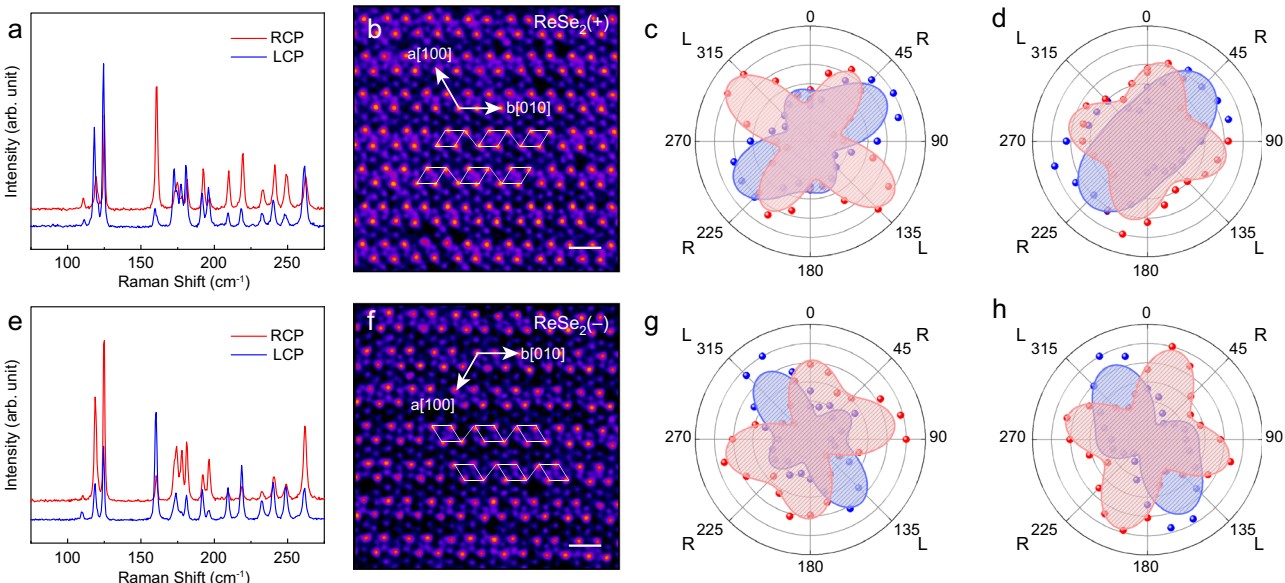

**Fig. 4 Circularly polarized Raman spectra of ReSe₂ with different vertical orientations. a** Circularly polarized Raman spectra of ReSe$_2$ (+) orientation ($E_L$ = 2.33 eV); **b** STEM-ADF image of ReSe$_2$ (+), scale bar: 0.5 nm.; **c, d** Polar plots of normalized Raman intensities of modes at 119 cm$^{-1}$ **c** and 162 cm$^{-1}$ **d** for 1.96 (red) and 2.33 eV (blue) excitation energies; **e** Circularly polarized Raman spectra of ReSe$_2$ (−) ($E_L$ = 2.33 eV); **f** STEM-ADF image of ReSe$_2$ (−); **g, h** Polar plots of normalized Raman intensities of modes at 119 cm$^{-1}$ **g** and 162 cm$^{-1}$ **h** for 1.96 (red) and 2.33 eV (blue) excitation energies.

approximation with norm-conserving pseudopotentials. A fine Monkhorst-Pack grid of 9 × 9 × 1 and 5 × 5 × 1 is used to sample the first Brillouin zone, respectively. We use the energy cut-off of 150 Ry to expand the wave functions with $10^{-13}$ Ry energy convergence threshold for self-consistency. A vacuum region of 20 Angstroms in the z-direction is used to avoid spurious interactions between two monolayers. The geometry is optimized using the quasi-newton algorithm, until none of the residual Hellmann–Feynman forces exceeds $10^{-5}$ Ry/Bohr and all components of stress tensor are less than 0.01 kbar. At last, we get the matrix elements on a much fine grid of 45 × 45 × 1 which is dense enough to achieve satisfactory results.

**Reporting summary**. Further information on research design is available in the Nature Research Reporting Summary linked to this article.

## Data availability
Most data generated or analyzed during this study are included in this published article or the Supplementary Materials. All data are available from the authors upon reasonable request.

## Code availability
All codes used for analysis of this study are available from the corresponding authors upon reasonable request.

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

## Acknowledgements

This work was financially supported by the Ministry of Science and Technology of China (Grant Nos. 2018YFA0703502, L.T.; 2016YFA0200104, J.Z.; and 2017YFA0206301, T.Y.), the National Natural Science Foundation of China (Grant Nos. 52021006, J.Z. and L.T.; 51720105003, J.Z.; 21790052, J.Z.; 21974004, L.T. and 52031014, Z.Z. and T.Y.), the Strategic Priority Research Program of CAS (Grant No. XDB36030100, J.Z.), and the Beijing National Laboratory for Molecular Sciences (Grant No. BNLMS-CXTD-202001, J.Z.). The simulation work was carried out at National Supercomputer Center in Tianjin, China, and the calculations were performed on TianHe-1(A).

## Author contributions

S.Z. prepared the specimens and performed all the Raman scattering. J.H. performed the theoretical calculations. Y.Y. and S.W. collected the STEM-ADF data. T.Y. and Z.Z. supervised all the theoretical calculations. L.T. and J.Z. supervised all the experiments and data collection. All authors contributed to the discussion of data and the writing of the manuscript.

## Competing interests

The authors declare no competing interests.
