## [Peer Review File · Nature Communications]

Reviewers' Comments:

Reviewer #1:

Remarks to the Author:

In this work, the authors use Raman spectroscopy to probe electron-phonon interaction in atomically thin layers of ReS₂ and ReSe₂, which possess anisotropic in-plane properties due to their low symmetry.

The authors published similar results earlier (Ref 23) as "Anomalous Polarized Raman Scattering and Large Circular Intensity Differential in Layered Triclinic ReS₂" ACS Nano 2017, 11, 10366–10372 (at <https://pubs.acs.org/doi/10.1021/acsnano.7b05321>)

They attributed the CID to "The origin of CID and the anomalous behavior in polarized Raman scattering were attributed to the appearance of nonzero off-diagonal Raman tensor elements and the phase factor owing to optical birefringence" in the abstract.

They should discuss if their interpretation has changed. Otherwise, how is this study different from Ref 23?

Raman intensities should include the excitonic effects as done in Ref 12. Miranda, H. P. C. et al. Quantum interference effects in resonant Raman spectroscopy of single- and triple-layer MoTe₂ from first-principles. Nano Lett. 17, 2381-2388 (2017).

There are multiple factors of which the effect on the circular intensity difference (CID) should be clearly explained and distinguished from the rest to clearly communicate the major claims. Otherwise, they are ambiguous:

1) Excitonic effects are excitation wavelength-dependent and they have an effect on the absolute and relative intensities of the Raman modes,

2) Chirality of the material as given on Line 48 "the two distinct structures of ReX₂ can not overlap by any rotation about the axis perpendicular to the 2D plane, leading to 2D enantiomers in analogue to molecular enantiomers", This effect does not give rise to any circular intensity difference (CID) without quantum interference.

3) It is claimed that quantum interference causes the relative intensities of the Raman modes to differ when the excitation wavelength is changed.

4) Quantum interference and chirality results in different anisotropic behavior for changing wavelength.

Even though the major results are convincing, all these factors listed above make the paper confusing.

Other comments:

Line 23: "However, it can only be observed in few materials under certain excitation configurations." It would be good to elaborate on this sentence.

Did the authors consider the effect of interference by the multiple surfaces of the thin fused silica substrates (170 μm in thickness)?

Line 222: "The ReS₂ and ReSe₂ flakes were transferred on SiNx grid by a polymer assisted approach." Details should be given so that other can reproduce the same study.

Line 117. T_n, e_s and e_i are not defined.

Reviewer #2:

Remarks to the Author:

The authors present chiral Raman scattering in ReS₂, which is more detailed extension of their study referenced under Ref. 23. The novelty of the manuscript is related to the studies of interference effect, which is responsible for the observed difference between RCP and LCP scattering signal. As such, the manuscript can be considered for publication, however several points should be addressed before its further processing.

In the reviewer opinion the order of the presentation is not the best possible. The interference and its explanation is the key point, the main conclusion of the manuscript. Therefore, for the sake of the presentation, the section on two distinct ReS₂ enantiomers should rather precede the conclusion. The discussion on the interference is based on the difference between RCP and LCP, which is a particular result of the characterization measurement presented in Figure 3. The authors should also stress more on the novelty of the section as compared to the conclusion of their previous work (Ref 23). Moreover the relevance of the results in ReSe₂ for the integrity of the whole message is not clear. Although the results are interesting, they seem to be outside the main line of the manuscript.

At top of those general comments some other issues should be addressed.

The authors properly point out that reports on resonance Raman scattering in two-dimensional materials are scarce. Therefore a reference to the paper by M. Osiekowicz et al., Sc. Reports 11, 924 (2021) claiming observations of the effect in GaSe and InSe would be helpful.

There is some confusion related to the section between lines 70 and 88:

The authors claim that the I-VI modes „are attributed to the vibrations of Re-Re bonds” (line 76). It is important to refer to some works, which confirm the statement. In fact, the eigenvectors of the modes are presented in Supporting information to A.McCreary et al, Nano Lett. 2017, 17, 10, 5897–5907 and it can be seen in the Figure S2 of the paper that both Re-Re and Re-S bonds are involved in the vibrations.

Technical quality of Fig.1 must be improved. There are no scale caption in the inset. The insets should not overlap with the main figures. The panel (d) shows most likely the CID defined in the text and not just the difference between intensities.

There are seven CIDs listed in lines 83-84 which are apparently related to six modes. Moreover just one of the values is negative, while both modes IV and VI are claimed to have negative CID.

The reference to “the rotation angle of the quarter wave plate” line 156 is rather technical. To make the statement more clear for the broader readership, some more details are needed, such as a scheme of experimental setup with crucial optical elements. Also “the polarization states of excitation” means presumably linear polarization of excitation light? The description also assume room temperature of the measurements, which is not actually clearly presented. Supplemental information should be updated with this respect.

Last but not least, it is strongly recommended to screen the manuscript from linguistic point of view. There are some typos, which definitely should be corrected such as:

Line 82 “ mode III and VI exhibits”

Line 134-135 “phase angles ... are ... parallel or antiparallel”

Line 136 “ .. distribution... are”

Line 39 of SI “In this session”.

Reviewer #3:

Remarks to the Author:

In this work, Zhang et. al reports the observation of a chiral Raman response in triclinic Rhenium

dichalcogenides (ReS₂ and ReSe₂) using left/right circularly polarized light as an excitation. They observe a large circular intensity difference that depends on the mode and excitation laser energy. They carry out a theoretical analysis of amplitudes and phase of the induced electric dipole when excited by left or right circularly polarized light and they show that, on the contrary to the amplitudes, the phases are different for left and right circular polarization and for different k_s . The difference in Raman scattering intensity measured, matches with the calculations when a full quantum interference effect is considered in the calculations.

The investigation seems to have been carried out with care and the theoretical methods are correctly related to the experimental results. Measuring quantum interference effects using such a readily available technique, as Raman spectroscopy in a layered material is noteworthy and definitely of great impact in the point of view of fundamental physics, Raman spectroscopy, and 2D materials. With that being said, I would like to suggest some improvements.

First, some suggestions that require more time and effort to address:

1 - What would be the results if a laser with sub-bandgap energy would be used for measuring the Raman spectra of the studied materials? Do the authors expected to observe such effects when an absorption to the conduction band and emission to the valence band is not taking place? Since the energy bandgap of ReS₂ and ReSe₂ is in the visible range, especially ReS₂, such experiment would not be so hard to be performed and will improve the work as well as more strongly corroborate the quantum interference interpretation.

2 - During theoretical calculations, the authors only considered band-band transitions. While just this picture seems to be enough to explain the results, I wonder how many bodies and excitonic effects could influence the results. Especially considering the large excitonic binding energy due to confinement effects. Can the authors discuss this more?

Some minor suggestions that will improve the reading experience.

1 - In Figures 1a and b - the color of the atoms is hard to distinguish, especially for Re(+) and Re(-). Can the authors choose more contrasting colors?

2 - Also in Figures 1a and b- please add the crystallographic axes.

3 - In supplementary figure S2 the yellow mark mention on line 97 of the main manuscript is almost invisible making it hard to see. Please correct this.

4 - In Figures 1c and d the authors present the experimental results for 2.33eV and 1.96eV excitation respectively while in Figures 2a and b they present the theoretical calculation for 1.96eV and 2.33eV respectively. The order in which the results are present is inverted comparing figures 1 to 2 making it a bit more confusing to compare. Can you swap the order in which either of the results is presented? Either 2.33 with 1.96 in figure 1 or 1.96 with 2.33 in figure 2

5 - Line 98 - I didn't understand how it is possible to well distinguish the 6 Raman-active modes from the 3 IR-active from the phonon dispersion due to inversion symmetry.

6 - In line 209 the authors conclude that "quantum interference can lead to a pronounced chiral response of Raman scattering in achiral materials". This statement is not clear to me. Can the authors explain this better or change this conclusion?

List of Changes

1. According to the comments of Reviewer #2, the order and relevant discussion of Fig.2 and Fig.3 are changed. In the revised manuscript, we first discussed the different chiral Raman response in ReS_2 (+) and ReS_2 (-) as shown in Fig.2, and gave the explanation based on quantum interference in Fig.3.
2. According to the comments of Reviewers, the content and quantity of Fig.1 has been improved, including the optical setup, the visualization of the vibrational modes, and the order of description for the 1.96 eV and 2.33 eV excitation, and the relevant discussion was added in the manuscript in line 61, 68 and 73.
3. The crystal structure of ReX_2 in original Fig. 1 was removed to supplementary information as Fig. S1 with more contrasting colors.
4. The novelty of this work has been emphasized according to the comments of Reviewer #1 and Reviewer #2 in line 47.
5. The CID values of ML- ReS_2 have been recalculated and shown in Line 82.
6. T_n , e_i and e_s are defined on line 148.
7. In line 194, the relevance of the results in monolayer ReSe_2 to this work was stressed.
8. Some ambiguous sentences and typos are revised in the manuscripts, including line 22 (Reviewer #1), line 78 (Reviewer #2), line 137 (Reviewer #3), line 166 (Reviewer #2), line 168 (Reviewer #2), line 222 (Reviewer #3) and line 43 in supplementary information (Reviewer #2).
9. In the Method section, the details of polymer assisted transfer method and the experimental condition are added.

10. The reference 'Osiekowicz, M. *et al.* Resonance and antiresonance in Raman scattering in GaSe and InSe crystals. *Sci. Rep.***11**, 924, (2021)' was cited as ref 11 according to comment of reviewer #2.

11. The yellow mark of Fig. S3 has been corrected according to the suggestion of Reviewer #3.

REVIEWER COMMENTS

Reviewer #1:

Question 1:

In this work, the authors use Raman spectroscopy to probe electron-phonon interaction in atomically thin layers of ReS₂ and ReSe₂, which possess anisotropic in-plane properties due to their low symmetry. The authors published similar results earlier (Ref 23) as "Anomalous Polarized Raman Scattering and Large Circular Intensity Differential in Layered Triclinic ReS₂" ACS Nano 2017, 11, 10366-10372 (<https://pubs.acs.org/doi/10.1021/acsnano.7b05321>).

They attributed the CID to "The origin of CID and the anomalous behavior in polarized Raman scattering were attributed to the appearance of nonzero off-diagonal Raman tensor elements and the phase factor owing to optical birefringence" in the abstract. They should discuss if their interpretation has changed. Otherwise, how is this study different from Ref 23.

Our response:

Thank you very much for your constructive comments. This is the exact reason why we studied *monolayer* ReS₂ in this work. In Ref. 23, we reported the chiral Raman responses of ReS₂ flakes that are as thick as several tens of nanometers (e. g. 32 nm, 80 nm), which can be considered as *bulk material*, where the optical effects, including birefringence and dichroism, can be prominent (*Nano Res.* 2018, 11, 3154-3163; *PRL* 2016, 116, 127401; *ACS Photonics* 2018, 5, 2509-2515).

Such optical effects induce either the modulation of the incident and scattered electric field (*Small, 2016, 12, 2627-2633*), or the change of complex Raman tensor (*PRL 2016, 116, 127401*). The chiral Raman scattering can thus be explained by the optical effects as discussed in Ref 23. It should be noted that the thickness-dependent CID values in Ref 23 can only be explained by the optical effects, because the electronic band structure and phonon dispersion are considered the same for bulk materials (*Nano Lett. 2017, 17, 5187-5192*). In order to find out if the optical effects are the only origin of the chiral Raman scattering or not, we studied monolayer ReS₂ (~0.7 nm thick, in this work), which has negligible phase difference of the ordinary and extraordinary light ($\Delta n = 0.037 \pm 0.009$ for ~520 nm, *ACS Photonics 2017, 4, 3023-3030*). Significant CID values are also found for monolayer ReS₂, which can not be attributed to the anisotropic optical environment, but can be attributed to the quantum interference effect in the optical transition and electron-phonon interaction processes.

We should note here that the quantum interference may also exist in bulk materials, which has not been observed since the optical effects already played such a dominant role that the quantum interference was overwhelmed. We also note that the statement in Ref 23: “The origin of CID and the anomalous behavior in polarized Raman scattering were attributed to the appearance of nonzero off-diagonal Raman tensor elements and the phase factor owing to optical birefringence” is correct. We calculated the Raman tensor for monolayer ReS₂ by taking the quantum interference into account, and nonzero off-diagonal elements also appear.

In order to emphasize the novelty of this work, we revised our manuscript in line 47 as: ‘... It has been reported that chiral Raman scattering arose in ReS₂ flakes of tens of nanometers in thickness, where the anisotropic optical effects play a prominent role. However, the underlying fundamental

interactions between photons/electrons/phonons during the chiral Raman scattering process, apart from the optical effects, have remained unexplored...’ and in line 86 as ‘...It should also be noted that owing to the monolayer thickness (~ 1 nm), the optical birefringence can be negligible...’

Question 2:

Raman intensities should include the excitonic effects as done in Ref 12. Miranda, H. P. C. *et al.* Quantum interference effects in resonant Raman spectroscopy of single- and triple-layer MoTe₂ from first-principles. *Nano Lett.* 17, 2381-2388 (2017).

Our response:

Thanks for your constructive comments. This is a good point. As done in Ref 12, Miranda et al. did include the excitonic effects in the study of quantum interference effect in resonant Raman spectroscopy of MoTe₂. We agree that, indeed, the excitonic effects are also important in anisotropic ReS₂ and ReSe₂ monolayer, and inclusion of such an effect will improve the Raman intensity calculation. However, we show here that it would not affect the main conclusion of this manuscript if the excitonic effect were taken into account, in terms of the following arguments:

Our calculations show that optical absorption in the studied systems doesn't make difference between RCP and LCP for the same laser energy, where we did not consider the excitonic effect. Since optical absorption is directly relevant to the squared absolute value of the electron-photon matrix element, the electron-photon matrix element M_{opt} in our system should share the common absolute values for both RCP and LCP at the same laser energy, even with the excitonic effect considered. Furthermore, from the Eq. (4) in the computation method of the manuscript, the common absolute value M_{opt} will result in the same Raman intensity (no CID) for both the LCP

and RCP, as long as the quantum interference effect is not considered, as discussed in more details in the manuscript. So the quantum interference effect, as we claimed, is the essential reason for the chiral Raman Scattering. Given the challenge and difficulty of incorporating exciton effect into our calculation, we will keep working on the development of our code to take care of the excitonic effect in the future work. But in the present work, we believe that it is sufficient to demonstrate this point of view, even without the excitonic effect considered at the moment.

Question 3:

There are multiple factors of which the effect on the circular intensity difference (CID) should be clearly explained and distinguished from the rest to clearly communicate the major claims.

Otherwise, they are ambiguous:

- 1) Excitonic effects are excitation wavelength-dependent and they have an effect on the absolute and relative intensities of the Raman modes.
- 2) Chirality of the material as given on Line 48 "the two distinct structures of ReX_2 can not overlap by any rotation about the axis perpendicular to the 2D plane, leading to 2D enantiomers in analogue to molecular enantiomers". This effect does not give rise to any circular intensity difference (CID) without quantum interference.
- 3) It is claimed that quantum interference causes the relative intensities of the Raman modes to differ when the excitation wavelength is changed.
- 4) Quantum interference and chirality results in different anisotropic behavior for changing wavelength.

Even though the major results are convincing, all these factors listed above make the paper

confusing.

Our response:

Thank you for your constructive comment. The major finding of this work is the quantum interference induced chiral Raman scattering. Other factors, such as the wavelength dependence and the discussion of 2D enantiomers, are to support this conclusion. However, as the reviewer pointed out, some of them may not be clearly explained and led to confusion in the original manuscript. In order to make the frame of this work more clear, we revised the main text by changing the order and the relevant discussion of Fig.2 and Fig. 3. We first discussed the different chiral Raman response in ReS₂ (+) and ReS₂ (-) as shown in Fig.2, and gave the explanation based on quantum interference in Fig.3. In order to show that the quantum interference effect generally exists in resonance Raman scattering process, at least in triclinic crystal systems, the results for ReSe₂ are exhibited as Fig.4 in main text. The main text was revised to stress the relevance of the results in ReSe₂ in line 194 as ‘...Although the quantum interference effect in Raman scattering has been reported in other 2D materials, this is the first observation of quantum interference leading to pronounced chiral Raman response. Furthermore, it can be expected that quantum interference exists in Raman scattering of a broader frame of materials. We show that the chiral Raman scattering can also be observed in other triclinic 2D layered materials such as ReSe₂...’

Below we also reply to the Reviewer on the factors that were listed above, and made corresponding changes in the revised manuscript.

1) Excitonic effects are excitation wavelength-dependent and they have an effect on the absolute and relative intensities of the Raman modes.

Reply:

Excitonic effects play an important role in Raman scattering, and affect the intensities of Raman modes due to the different electron/exciton-phonon couplings. In this work, we show the chiral Raman scattering for excitation wavelengths of 532 nm and 633 nm in order to support the role played by the quantum interference, which exists as long as optical transition occurs between valence and conduction bands. We did not mean to compare Raman intensities between the two excitation laser energies due to the excitonic effects, but focused on comparing Raman spectra between the LCP and RCP excitation of the same laser energy. As in our reply to the pervious question, the excitonic effect changes but the absolute value of intensity, not the relative difference due to circular polarizations, so we think that it should not affect the main conclusion of the manuscript even with excitonic effects taken into account.

2) Chirality of the material as given on Line 48 "the two distinct structures of ReX_2 can not overlap by any rotation about the axis perpendicular to the 2D plane, leading to 2D enantiomers in analogue to molecular enantiomers". This effect does not give rise to any circular intensity difference (CID) without quantum interference.

Reply:

Indeed, it is the quantum interference that leads to the CID in monolayer ReX_2 . The fact that the two distinct structures of ReX_2 can not overlap by any rotation about the axis perpendicular to the 2D plane, which we define as 2D enantiomers, is a supportive finding to the chiral Raman scattering. ReX_2 belongs to C_i point group, which is an achiral point group. However, owing to the confinement of the 2D plane, ReX_2 response chirally to the circularly polarized Raman scattering

through the quantum interference effect. Furthermore, the two structures (ReS₂ (+) and ReS₂ (-)) have enantiomorphous Brillouin zone and band structure, and for each Raman scattering process with one photo-excited electron involved, the phases of the induced electric dipoles for ReS₂ (+) and ReS₂ (-) have opposite signs, leading to the opposite Raman response, that is, the intensities of each Raman mode $I_{(ReS_2(+))_LCP}$ equals to $I_{(ReS_2(-))_RCP}$.

3) It is claimed that quantum interference causes the relative intensities of the Raman modes to differ when the excitation wavelength is changed.

Reply:

In this work, the different intensities of the same Raman mode excited by LCP and RCP were explained by the quantum interference. However, the physical picture of quantum interference should be universal for resonance Raman scattering. If the reviewer refers relative intensities to different Raman modes, such as the intensity ratio of mode I to mode II, it should differ because of the different optical absorption and the consequential electron-phonon coupling (*Nano Lett.* 2016, 16, 4, 2260-2267). If the relative intensities refer to the ratio of the same Raman mode excited by LCP and RCP, such as I_{LCP}/I_{RCP} for the same excitation laser, this can be explained by the quantum interference.

In order to avoid confusion, we removed the relevant sentence “This can be easily understood from the different optical transitions and electron-phonon...”, and discuss the optical transitions in the discussion of quantum interference.

4) Quantum interference and chirality results in different anisotropic behavior for changing

wavelength.

Reply:

In this work, the quantum interference and chirality are not relevant to the anisotropic behavior of ReS_2 , which is only determined by its triclinic crystalline structure. The anisotropic behavior of the Raman scattering in monolayer ReS_2 with different excitation wavelengths could be changed due to the change of initial and final states in optical transition processes, which have also been proven by some references (*ACS Nano* 2016, 10, 2, 2752–2760, *Nano Lett.* 2017, 17, 10, 5897–5907). The anisotropic behavior of triclinic crystal is also vertical orientation dependent (*Nano Lett.* 2016, 16, 2, 1381–1386). We also measured the linear polarization dependence of ReS_2 with different vertical orientations as shown below.

Figure.R1 Polar Plots of ReS_2 (+) and ReS_2 (-) under $Z(YY)Z$ configuration. The excitation photon energy is 2.33 eV.

Question 4:

Line 23: "However, it can only be observed in few materials under certain excitation configurations." It would be good to elaborate on this sentence.

Our response:

Thank you very much for your suggestion. From the findings in this work, we conclude that quantum interference generally exists in resonance Raman scattering process. However, it is difficult to observe experimentally since it is usually accompanied by more prominent effects, such as resonant enhancement and optical effects. Up to now, the quantum interference has only been reported in very few systems, such as doped graphene (Ref 12) and few-layer MoTe₂ (Ref 13). In electrostatically doped graphene (Ref 12), if all quantum pathways are allowed, the destructive interference leads to a weak overall Raman signal, but when some of the Raman scattering pathways are gate-tunable blocked, the one phonon Raman intensity increasing dramatically. In triple-layer MoTe₂ (Ref 13), with increasing excitation energy, the intensity of A'_1 vibrational modes (a and b peaks for Davydov splitting) will be inversed. Hence, in order to observe the effect of quantum interference, other effects should be suppressed, for example, by electrical gating. In our work, we chose monolayer ReS₂, for which the optical effects are negligible, and used circular polarization for excitation to observe the effect of quantum interference.

We revised this sentence in line 22 to ‘However, it *has only been* observed in few materials under certain excitation configurations.

Question 5:

Did the authors consider the effect of interference by the multiple surfaces of the thin fused silica substrates (170 μm in thickness)?

Our response :

Thank you very much for your constructive comments. Optical interference can enhance the

Raman intensity due to multiple reflection when the thickness of the 2D materials or the substrate is comparable with the incident or scattered wavelength. The interference is prominent for SiO₂/Si substrates. In this work, we chose fused silica substrate (170 μm in thickness) SiO₂/Si in order to minimize the effect of interference. The interference enhancement can be calculated according to the following.

The schematic of multiple reflection is shown as below:

Figure R2. The interference effect in Raman scattering of ReS₂ samples on fused silica substrate.

The net enhancement of incident light ($F_{ex}(x)$) at depth x measured from the ReS₂ surface is given by

$$F_{ex}(x) = t_{01} \frac{(1 + r_{12}r_{23}e^{-2i\beta_2^{ex}x})e^{-i\beta_x^{ex}x} + (r_{12} + r_{23}e^{-2i\beta_2^{ex}x})e^{-i(2\beta_1^{ex} - \beta_x^{ex})x}}{1 + r_{12}r_{23}e^{-2i\beta_2^{ex}x} + (r_{12} + r_{23}e^{-2i\beta_2^{ex}x})r_{01}e^{-2i\beta_1^{ex}x}}$$

where $t_{ij} = 2n_i/(n_i + n_j)$, and $r_{ij} = (n_i - n_j)/(n_i + n_j)$ according to the Fresnel's equation at the surface from i to j , including air (0), ReS₂ (1), SiO₂ (2) and air (3). $\beta_x^{ex} = 2\pi x n_1/\lambda_{ex}$ and $\beta_i^{ex} = 2\pi d_i n_i/\lambda_{ex}$ are the phase factor, where λ_{ex} is the excitation wavelength, the d_i is the thickness of medium i .

The scattering enhancement is given by

$$F_{sc}(x) = t_{10} \frac{(1 + r_{12}r_{23}e^{-2i\beta_2^{sc}x})e^{-i\beta_x^{sc}x} + (r_{12} + r_{23}e^{-2i\beta_2^{sc}x})e^{-i(2\beta_1^{sc} - \beta_x^{sc})x}}{1 + r_{12}r_{23}e^{-2i\beta_2^{sc}x} + (r_{12} + r_{23}e^{-2i\beta_2^{sc}x})r_{01}e^{-2i\beta_1^{sc}x}}$$

The total enhancement is

$$F = N \int_0^{d_1} |F_{ex}(x)F_{sc}(x)|^2 dx$$

Since ReS₂ is anisotropic, and all the Raman tensor elements have real part and imaginary part according to our theoretical calculations, the anisotropic interference effect may influence the relative Raman intensity excited by RCP and LCP, but is much less than the effect of quantum interference as shown below.

Here, we calculated the enhancement factor along the *b*-axis ($F_{parallel}$) and perpendicular to the *b*-axis (F_{cross}). With 532.5 nm (2.33 eV) excitation wavelength, the scattered wavelength of Mode I ~Mode VI locate at around 537 nm, and the refractive index (n and k) is obtained from reference (*Nanoscale*, 2019, 11, 20199). The ratio of the enhancement factor $F_{parallel}/F_{cross}$ is 1.13, and $(F_{parallel} - F_{cross})/(F_{parallel} + F_{cross})$ is around 0.06 (the maximum value), which is much smaller than the CID values, so the optical interference is not the key factor of the chiral Raman response.

Question 6:

Line 222: "The ReS₂ and ReSe₂ flakes were transferred on SiN_x grid by a polymer assisted approach." Details should be given so that other can reproduce the same study.

Our response:

Thank you for your kind suggestion. The transfer approach has been given in the Method section in the revised manuscript on line 234. '...PPC/PDMS stamps were used to pick the ReS₂ flakes from the substrates, and then covered onto the SiN_x grids using a transfer stage. The temperature was raised to 110 °C until the stamps and the grids were well contacted, and then the grids and the stamps were placed in acetone for 24 h at room temperature to remove the PPC...'

Question 7:

Line 117. T_n , e_s and e_i are not defined.

Our response:

Thank you very much for pointing this out. We revised the main text in line 146 "...In the simplest scenario, 4 electrons at two k points are assumed to be involved in the Raman scattering process, for each scattering channel $T_n = e_s * R_n * e_i$, in which R_n is the Raman tensor, e_i and e_s are Jones vectors of incident and scattering polarization, respectively..."

Reviewer #2:**Question 1:**

The authors present chiral Raman scattering in ReS_2 , which is more detailed extension of their study referenced under Ref. 23. The novelty of the manuscript is related to the studies of interference effect, which is responsible for the observed difference between RCP and LCP scattering signal. As such, the manuscript can be considered for publication, however several points should be addressed before its further processing.

Our response:

We thank the reviewer for the positive comments! We have carefully revised our manuscript accordingly, and replied to the comments point-by-point below.

In the reviewer opinion the order of the presentation is not the best possible. The interference and its explanation are the key points, the main conclusion of the manuscript. Therefore, for the sake of the presentation, the section on two distinct ReS_2 enantiomers should rather precede the

conclusion. The discussion on the interference is based on the difference between RCP and LCP, which is a particular result of the characterization measurement presented in Figure 3. The authors should also stress more on the novelty of the section as compared to the conclusion of their previous work (Ref 23). Moreover, the relevance of the results in ReSe₂ for the integrity of the whole message is not clear. Although the results are interesting, they seem to be outside the main line of the manuscript.

Our response:

Thank you very much for your constructive comments. According to your suggestions, we revised the main text by changing the order and the relevant discussion of Fig.2 and Fig. 3. We first discussed the different chiral Raman response in ReS₂ (+) and ReS₂ (-) as shown in Fig.2, and gave the explanation based on quantum interference in Fig.3. In order to show that the quantum interference effect generally exists in resonance Raman scattering process, at least in triclinic crystal systems, the results for ReSe₂ are exhibited as Fig.4 in main text. The main text was revised to stress the relevance of the results in ReSe₂ in line 193 as ‘...Although the quantum interference effect in Raman scattering has been reported in other 2D materials, this is the first observation of quantum interference leading to pronounced chiral Raman response. Furthermore, it can be expected that quantum interference exists in Raman scattering of a broader frame of materials. We show that the chiral Raman scattering can also be observed in other triclinic 2D layered materials such as ReSe₂...’

The novelty of this work and the relevance of this work to Ref 23 are also asked by Reviewer 1, and we emphasize that this is the exact reason why we studied *monolayer* ReS₂ in this work. We

quote here the same reply below:

In Ref. 23, we reported the chiral Raman responses of ReS₂ flakes that are thick as over several tens of nanometers (e. g. 32 nm, 80 nm), which can be considered as bulk material, where the optical effects, including birefringence and dichroism, can be prominent (*Nano Res.* 2018, 11, 3154-3163; *PRL* 2016, 116, 127401; *ACS Photonics* 2018, 5, 2509-2515). Such optical effects induce either the modulation of the incident and scattered electric field (*Small*, 2016, 12, 2627-2633), or the change of complex Raman tensor (*PRL* 2016, 116, 127401). The chiral Raman scattering can thus be explained by the optical effects as discussed in Ref 23. It should be noted that the thickness-dependent CID values in Ref 23 can only be explained by the optical effects, because the electronic band structure and phonon dispersion are the same for bulk materials (*Nano Lett.* 2017, 17, 5187-5192). In order to find out if the optical effects are the only origin of the chiral Raman scattering or not, we studied monolayer ReS₂ (~ 0.7 nm thick, in this work), which has negligible phase difference of the ordinary and extraordinary light ($\Delta n = 0.037 \pm 0.009$ for ~520 nm, *ACS Photonics* 2017, 4, 3023–3030). Significant CID values are also found for monolayer ReS₂, which can not be attributed to the anisotropic optical environment, but attributed to the quantum interference effect in the optical transition and electron-phonon interaction processes.

We should note here that the quantum interference may also exist in bulk materials, which has not been observed since the optical effects already played such a dominant role that the quantum interference was overwhelmed. We also note that the statement in Ref 23: “The origin of CID and the anomalous behavior in polarized Raman scattering were attributed to the appearance of nonzero off-diagonal Raman tensor elements and the phase factor owing to optical birefringence”

is also correct. We calculated the Raman tensor for monolayer ReS₂ by taking the quantum interference into account, and nonzero off-diagonal elements also appear.

In order to emphasize the novelty of this work, we revised our manuscript on line 47 as: ‘...It has been reported that chiral Raman scattering arose in ReS₂ flakes of tens of nanometers in thickness, where the anisotropic optical effects play a prominent role. However, the underlying fundamental interactions between photons/electrons/phonons during the chiral Raman scattering process, apart from the optical effects, have remained unexplored...’ and on line 87 as ‘...It should also be noted that owing to the monolayer thickness (~ 1 nm), the optical birefringence can be negligible...’

Question 2:

The authors properly point out that reports on resonance Raman scattering in two-dimensional materials are scarce. Therefore, a reference to the paper by M. Osiekowicz et al., Sci. Reports 11, 924 (2021) claiming observations of the effect in GaSe and InSe would be helpful.

Our response:

Thank you very much for your kind advice. This paper is very helpful for understanding the resonance and anti-resonance effect in Raman scattering, and this paper have been added in the revised manuscript as Ref 11.

Question 3:

There is some confusion related to the section between lines 70 and 88:

The authors claim that the I-VI modes are attributed to the vibrations of Re-Re bonds” (line 76). It is important to refer to some works, which confirm the statement. In fact, the eigenvectors of the

modes are presented in Supporting information to A. McCreary et al, Nano Lett. 2017, 17, 10, 5897–5907 and it can be seen in the Figure S2 of the paper that both Re-Re and Re-S bonds are involved in the vibrations.

Our response:

Thank you very much for your constructive comments! We have read the suggested paper carefully, and cited them in revised manuscript as Ref 24 and 25. Owing to the different reduced mass of Re atom and S atom, the low frequency vibrational modes mostly originate from the contribution of Re atoms, however, there is indeed still some minor contribution of S atoms, which increases from low to high vibrational frequencies. We revised the sentence in line 73 as “...The eigenvectors of the Raman-active vibrational modes (marked as mode I-VI) were calculated using density functional theory (DFT) and are shown in Fig. 1b...”. Meanwhile, we have added the visualization of the vibrational modes in Fig. 1b.

Question 4:

Technical quality of Fig.1 must be improved. There are no scale caption in the inset. The insets should not overlap with the main figures. The panel (d) shows most likely the CID defined in the text and not just the difference between intensities.

Our response:

Thank you very much for your constructive comments! We have improved the technical quality and contents of Fig.1 as below.

Fig.1

Figure 1. (a) Optical setup of the chiral Raman scattering measurements; (b) The eigenvectors of the Raman-active vibrational modes (I~VI) according to DFT calculations; (c, d) Circularly polarized Raman spectra for the Re vibrational modes of 1L ReS₂ ($E_L=1.96$ eV (c) and 2.33 eV (d)). The insets depict the difference between the Raman intensities of 1L ReS₂ excited by RCP and LCP.

The insets in Fig.1c and 1d are the intensity difference between RCP and LCP excitation. The CID value is defined as $(I_R - I_L)/(I_R + I_L)$, which are not depicted in Fig.1 because of the noise of baseline (due to very small denominators), so we use the intensity difference instead and calculated the CID values in main text according to the peak intensities fitted by Lorenz line (*J. Am. Chem. Soc.* 1973, 95, 2, 603).

Question 5:

There are seven CIDs listed in lines 83-84 which are apparently related to six modes. Moreover, just one of the values is negative, while both modes IV and VI are claimed to have negative CID.

Our response:

Thank you very much for pointing this out! There should be six CID values corresponding to the six modes. We apologize for putting a wrong series of CID values in the original manuscript. The CID values were recalculated and revised in the manuscript on line 82 as ‘...The circular intensity differential (CID, Δ) values, defined as $(I_R - I_L)/(I_R + I_L)$, are observed to be 0.49, 0.33, 0.18, -0.05, 0.27 and -0.16 for modes I-VI respectively, for 2.33 eV excitation...’

Question 6:

The reference to “the rotation angle of the quarter wave plate” line 156 is rather technical. To make the statement clear for the broader readership, some more details are needed, such as a scheme of experimental setup with crucial optical elements. Also “the polarization states of excitation” means presumably linear polarization of excitation light? The description also assumes room temperature of the measurements, which is not actually clearly presented. Supplemental information should be updated with this respect.

Our response:

Thank you very much for your constructive comments! The optical setup has been added as Fig. 1a in the revised main text, which is also shown below.

Fig.1

With the rotation of QWP, the polarization states for excitation can be changed from linear polarization to elliptical polarization and circular polarization. There is no analyzer in the pathway of the scattered light, so the scattered light with all the polarization states are collected. The measurements were performed at room temperature. The details of the measurement have been added in the Method section ‘...Raman spectra were measured using JY Horiba HR800 with 2.33 eV and 1.96 eV excitation energy at room temperature...’

Question 7:

Last but not least, it is strongly recommended to screen the manuscript from linguistic point of view. There are some typos, which definitely should be corrected such as:

Line 82 “mode III and VI exhibits”

Line 134-135 “phase angles are ... parallel or antiparallel”

Line 136 “ .. distribution... are”

Line 39 of SI “In this session”.

Our response:

Thank you very much for your kind comments! We have carefully revised the whole manuscript and double check from linguistic point of view. The typos have been corrected as below:

Line 77: ‘...When excited by 1.96 eV, only modes III (153 cm⁻¹) and VI (235 cm⁻¹) exhibit obvious chiral response but with opposite signs...’

Line 165: ‘...the phase angles of Raman tensors $R_{i,f,n,n'}(k)$ involving one electron are found to be either parallel or antiparallel to each other...’

Line 167: ‘...However, the k -resolved phase distribution of complex $T_{i,f,n,n'}(k) = e_s \cdot R_{i,f,n,n'}(k) \cdot e_i$ is different between LCP and RCP excitations...’

Line 43 of SI ‘in this section’

Reviewer #3:

In this work, Zhang et. al reports the observation of a chiral Raman response in triclinic Rhenium dichalcogenides (ReS₂ and ReSe₂) using left/right circularly polarized light as an excitation. They observe a large circular intensity difference that depends on the mode and excitation laser energy. They carry out a theoretical analysis of amplitudes and phase of the induced electric dipole when excited by left or right circularly polarized light and they show that, on the contrary to the amplitudes, the phases are different for left and right circular polarization and for different ks. The difference in Raman scattering intensity measured, matches with the calculations when a full quantum interference effect is considered in the calculations.

The investigation seems to have been carried out with care and the theoretical methods are correctly related to the experimental results. Measuring quantum interference effects using such a readily available technique, as Raman spectroscopy in a layered material is noteworthy and definitely of great impact in the point of view of fundamental physics, Raman spectroscopy, and 2D materials. With that being said, I would like to suggest some improvements.

First, some suggestions that require more time and effort to address:

Question1

What would be the results if a laser with sub-band gap energy would be used for measuring the Raman spectra of the studied materials? Do the authors expected to observe such effects when an absorption to the conduction band and emission to the valence band is not taking place? Since the energy band gap of ReS₂ and ReSe₂ is in the visible range, especially ReS₂, such experiment would not be so hard to be performed and will improve the work as well as more strongly corroborate the quantum interference interpretation.

Our response :

Thanks for your constructive suggestion. In this work, we calculated the quantum interference for resonant Raman scattering processes. For non-resonant Raman scattering, since there is no optical transition from the valence band to the conduction band, it is highly likely that quantum interference would not occur. So far we have not been able to experimentally perform non-resonant chiral Raman measurements as it needs different spectrometer. Ideally, a 1064 nm laser is suitable for non-resonant Raman measurements since the energy is below the band gap and it is a commercially available laser, but it needs to equip with a different CCD that has high quantum yield in the near-IR region. Nevertheless, we supplemented data of Raman scattering for 1.58 eV

(785 nm) excitation. The Raman spectra of monolayer ReS₂ excited by 1.58 eV (785 nm) excitation photon energy have been collected and shown Fig. R3.

Figure R3. (a) The photoluminescence spectra of monolayer ReS₂; (b) the chiral Raman scattering of monolayer ReS₂ excited by 1.58 eV photon energy; (c) CID values of the chiral Raman response.

The photoluminescence of monolayer ReS₂ was also measured and shown in Fig. R3(a), and the excitation photon energy of the chiral Raman scattering was shown with the dash line. The 1.58 eV excitation photon energy is slightly below the peak of the exciton luminescence of monolayer ReS₂. However, considering the peak broadening, 1.58 eV can still excite electrons transition from CB to VB and it can be considered as near-resonant Raman scattering. The chiral Raman scattering was shown in Fig. R3(b), and the mode III located in 153cm⁻¹ exhibit obviously different response excited by LCP and RCP, and the CID values have been calculated in Fig. R3(c).

The measurement of non-resonant Raman scattering and its comparison to resonant Raman

scattering would certainly be of great importance. We are working on searching appropriate materials with suitable band gap for such measurements.

Question2

During theoretical calculations, the authors only considered band-band transitions. While just this picture seems to be enough to explain the results, I wonder how many bodies and excitonic effects could influence the results. Especially considering the large excitonic binding energy due to confinement effects. Can the authors discuss this more?

Our response:

Thanks for your constructive comments. The excitonic effect is also concerned by Reviewer 1. We quote below our reply to Reviewer 1:

We agree that the excitonic effects are also important in anisotropic ReS₂ and ReSe₂ monolayer, and inclusion of such an effect will improve the Raman intensity calculation. However, we show here that it would not affect the main conclusion of this manuscript if the excitonic effect were taken into account, in terms of the following arguments:

Our calculations show that optical absorption in the studied systems doesn't make difference between RCP and LCP for the same laser energy, where we did not consider the excitonic effect. Since optical absorption is directly relevant to the squared absolute value of the electron-photon matrix element, the electron-photon matrix element M_{opt} in our system should share the common absolute values for both RCP and LCP at the same laser energy, even with the excitonic effect considered. Furthermore, from the Eq. (4) in the computation method of the manuscript, the common absolute value M_{opt} will result in the same Raman intensity (no CID) for both the LCP

and RCP, as long as the quantum interference effect is not considered, as discussed in more details in the manuscript. So the quantum interference effect, as we claimed, is the essential reason for the chiral Raman Scattering. Given the challenge and difficulty of incorporating exciton effect into our calculation, we will keep working on the development of our code to take care of the excitonic effect in the future work. But in the present work, we believe that it is sufficient to demonstrate this point of view, even without the excitonic effect considered at the moment.

Some minor suggestions that will improve the reading experience.

Question 1

In Figures 1a and b - the color of the atoms is hard to distinguish, especially for Re(+) and Re(-).

Can the authors choose more contrasting colors?

Our response:

Thank you very much for your kind advice! The schematics of three dimensional and two dimensional ReS₂ structures have been revised in Fig. S1 with more contrasting color, as shown below.

Figure S1. Schematics of three dimensional (a) and two dimensional (b) ReS₂ structure

Question2

Also in Figures 1a and b- please add the crystallographic axes.

Our response:

Thank you very much for your suggestion! The crystallographic axes have been added in Fig. S1.

Question 3

In supplementary figure S2 the yellow mark mention on line 97 of the main manuscript is almost invisible making it hard to see. Please correct this.

Our response:

Thank you very much for your advice! The yellow color in this Figure has been corrected in Fig. S3 in supplementary information as shown below.

Figure S3. Calculated electronic band structure (a) and phonon dispersion relation (b) of 1L ReS₂.

Question 4.

In Figures 1c and d the authors present the experimental results for 2.33eV and 1.96eV excitation respectively while in Figures 2a and b they present the theoretical calculation for 1.96eV and

2.33eV respectively. The order in which the results are present is inverted comparing figures 1 to 2 making it a bit more confusing to compare. Can you swap the order in which either of the results is presented? Either 2.33 with 1.96 in figure 1 or 1.96 with 2.33 in figure 2

Our response:

Thank you very much for your kind advice! The order of the experimental results for 2.33 eV and 1.96 eV excitation has been corrected, and is coincident with Figure 2 as shown below, and the order in Fig.3 is also changed.

Fig. 1

Question 5

Line 98 - I didn't understand how it is possible to well distinguish the 6 Raman-active modes from the 3 IR-active from the phonon dispersion due to inversion symmetry.

Our response :

Thank you very much for your comments. The previous description “due to inversion symmetry” was indeed ambiguous. The sentences have been revised in line 137 as ‘...There are 9 optical phonon modes in this region in which all the 6 gerade modes (A_g) are Raman-active while the other 3 ungerade modes (A_u) are infrared-active according to the character table of C_i point group...’

Question6

In line 209 the authors conclude that “quantum interference can lead to a pronounced chiral response of Raman scattering in achiral materials”. This statement is not clear to me. Can the authors explain this better or change this conclusion?

Our response :

Thank you very much for pointing this out! In this work, we measured the chiral Raman response of monolayer ReS_2 and ReSe_2 , which belong to the achiral C_i point group and $P(1)$ space group (*Physical Review B*, 2015, 92, 054110). The chiral Raman response of ReS_2 comes from the quantum interference between the different pathway of the Raman scattering process, which is not the same as Raman optical activity of chiral molecules (with typical CID values $\sim 10^{-3}$) (*Molecular Physics*, 1971, 20, 1111). In order to avoid confusion, the sentence has been revised on line 222 as “...quantum interference can lead to a pronounced chiral response of Raman scattering in materials...”.

Reviewers' Comments:

Reviewer #1:

The authors have made several improvements to the manuscript based on the points raised by the reviewers. However, I still have 2 major concerns:

1) This is about the mechanisms causing CID in bulk and monolayer ReS₂:

The authors noted the following on the last paragraph of page 5 in their earlier study Ref. 23 "Anomalous Polarized Raman Scattering and Large Circular Intensity Differential in Layered Triclinic ReS₂" ACS Nano 2017, 11, 10366–10372 (at <https://pubs.acs.org/doi/10.1021/acsnano.7b05321>)

The circular intensity differential is defined as the ratio Δ of the difference of the Raman scattered intensities in right and left circularly polarized incident light:

$$\Delta = \frac{I^R - I^L}{I^R + I^L} = \frac{2v(u + w) \sin \delta}{u^2 + 2v^2 + w^2} \quad (6)$$

The magnitude of the difference depends on the values of u , w , v , and $\sin \delta$, which differ for different Raman modes. The nonzero δ is only possible for anisotropic materials including orthorhombic, monoclinic, and triclinic crystals, and the nonzero $v(u + w)$ holds for the A_g mode of triclinic crystals and the monoclinic crystals.

Hence, it reads that a nonzero birefringence is required to observe circular intensity differential (CID). In the current work, they study the monolayer as opposed to the bulk thickness in Ref 23. They state that the monolayer thickness has negligible birefringence on Line 86. How can the monolayer have CID without significant birefringence? Can the authors explain if this is a contradiction? Does it mean that the quantum interference (QI) in monolayer is much larger than that in the thickness of 10s of nanometers? If so why?

I understand that for thick ReS₂, the magnitude of CID may depend on thickness mainly due to optical effects such as birefringence. However, I am not convinced that the origins of the CID observed in monolayer and thick ReS₂ are different. To me it looks like both monolayer and thick ReS₂ exhibit CID due to the same mechanism. The authors state in their response that CID in monolayer is mainly due to QI whereas the CID in bulk is due to optical birefringence.

I would expect birefringence and QI to enhance the CID rather than suppressing each other's effect on CID in bulk ReS₂. They say that QI may be overwhelmed in bulk. Why would that be the case?

I would expect a clear explanation and proof in the manuscript presented to the readers that would explain the mechanism or mechanisms causing CID in monolayer and bulk ReS₂. I would like to be sure that the results are attributed to the correct physical mechanism.

2) This is about the way Raman measurements were performed:

They write in the new version of the manuscript that "A quarter wave plate (QWP) is used to produce right-handed (RCP) or left-handed (LCP) circular polarization for excitation. The scattered light passes through the same QWP, and is collected without any analyzer." I believe this experimental configuration assumes that the Raman scattered light has the same polarization as the excitation light. However, is that not problematic given that the Raman tensor has off-diagonal elements? I would expect the scattered light to lose its circular polarized nature. They can test this by using an analyzer to see if their Raman scattered light is linearly polarized (parallel to the incident laser) after the quarter-wave plate as they assumed in the current version.

For instance, it is reported in Nano Lett. 2015, 15, 5667–5672 (at <https://pubs.acs.org/doi/abs/10.1021/acs.nanolett.5b00910>) that the Raman scattered light from monolayer ReS₂ has both parallel and perpendicular components with respect to the linearly polarized incident 532 nm laser.

Reviewer #2:

Remarks to the Author:

My comments have been successfully addressed and the quality of the manuscript improved in my opinion. The observation of quantum interference related effects in Raman scattering as previously stated is of interest to the broad community, and until now it was not very thoroughly investigated. The paper provides an important input confirming a diversity of effects in low-dimensional semiconductors.

I can recommend the manuscript in its present form for publication.

Reviewer #3:

Remarks to the Author:

The authors carefully addressed all the questions raised. The parts that were confusing and ambiguous were corrected. Regarding my suggestion of reproducing the experiment using a sub-bandgap excitation laser, I understand the experimental difficulty that arises from this, and I found the response given by the authors satisfactory. Nevertheless, it would be interesting seeing this experiment published in the future using another more convenient material.

Overall, the authors improved the manuscript, and I now found it suitable for publication.

Question 1

This is about the mechanisms causing CID in bulk and monolayer ReS₂:

The authors noted the following on the last paragraph of page 5 in their earlier study Ref. 23 "Anomalous Polarized Raman Scattering and Large Circular Intensity Differential in Layered Triclinic ReS₂" ACS Nano 2017, 11, 10366–10372 (at <https://pubs.acs.org/doi/10.1021/acsnano.7b05321>)

Hence, it reads that a nonzero birefringence is required to observe circular intensity differential (CID). In the current work, they study the monolayer as opposed to the bulk thickness in Ref 23. They state that the monolayer thickness has negligible birefringence on Line 86. How can the monolayer have CID without significant birefringence? Can the authors explain if this is a contradiction? Does it mean that the quantum interference (QI) in monolayer is much larger than that in the thickness of 10s of nanometers? If so why?

I understand that for thick ReS₂, the magnitude of CID may depend on thickness mainly due to optical effects such as birefringence. However, I am not convinced that the origins of the CID observed in monolayer and thick ReS₂ are different. To me it looks like both monolayer and thick ReS₂ exhibit CID due to the same mechanism. The authors state in their response that CID in monolayer is mainly due to QI whereas the CID in bulk is due to optical birefringence.

I would expect birefringence and QI to enhance the CID rather than suppressing each other's effect on CID in bulk ReS₂. They say that QI may be overwhelmed in bulk. Why would that be the case?

I would expect a clear explanation and proof in the manuscript presented to the readers that would explain the mechanism or mechanisms causing CID in monolayer and bulk ReS₂. I would like to be sure that the results are attributed to the correct physical mechanism.

Our response:

Thanks for your constructive comments. In the last few years, we have tried a lot to investigate the relevance of the circularly intensity differential (CID) to the birefringence and/or quantum interference (QI) effect, but only to find that there are still many difficulties to set up a complete model to incorporate all possible effects. To response to your reasonable doubt, here we make some more arguments and discussions on the possible mechanisms of CID in monolayer and bulk ReS₂.

First, we consider that the birefringence effect and the quantum interference effect, if both applicable to the understanding of CID, may not be able to fall into one single mechanism, given that the former is concerned with the phase difference in the real space (for instance, sample depth/thickness, refractive indices' difference), but the latter with the phase difference between matrix elements in the reciprocal k space.

Second, we can well separate the contribution to the CID of the birefringence effect from that of QI effect in *monolayer*. CID due to the birefringence effect can be evaluated roughly by $CID \propto \frac{2\pi d}{\lambda} \Delta n$ (phase retardation between the ordinary and extraordinary light), in which d is the sample thickness, Δn the difference of the refractive indices between ordinary and extraordinary light and λ the light wavelength. For example, in *monolayer* ReS₂, $d \sim 0.7$ nm, $\Delta n = 0.037 \pm 0.009$ for $\lambda \sim 520$ nm (data from *ACS Photonics* 2017, 4, 3023-3030), the phase retardation is approximately 0.0003, so the CID value induced by the birefringence effect is significantly smaller than both the experimental values ($\sim 0.05 - 0.54$) and the calculated QI-induced values ($\sim 0.04 - 0.60$), indicating a very minor role played by the anisotropic optical environment in *monolayer* ReS₂.

Third, sorry for the confusion by saying “QI may be overwhelmed in bulk” in our previous reply. We meant that birefringence and QI are complementary and both contribute to the CID in bulk ReS₂, as pointed out by the referee, and the birefringence can be strong due to the considerable thickness in bulk. But up to now, we still cannot separate the contribution of birefringence from that of QI to the CID in bulk by experiment, because it is really a difficult task to set up a quantitative model to calculate Raman scattering efficiency at different depths of crystal, as indicated as follows. The polarization vectors of incoming and outgoing electrical field at z depth inside sample can be described as (*Phys. Rev. B*, 2002, 65, 176403):

$$e_{in.out}(z, \omega) = \frac{\cos(\theta)e^{-\alpha_x z} \hat{x} + \sin(\theta)e^{-\alpha_y z} e^{\pm i\delta(z)} \hat{y}}{(\cos^2(\theta)e^{-2\alpha_x z} + \sin^2(\theta)e^{-2\alpha_y z})^{1/2}}$$

where $\alpha_{x,y}$ are absorption coefficients and the $+/-$ correspond to the incoming/outgoing electrical field. The polarization states depend on the penetration depth z and on the energy of incident and scattered light *via* the parameter δ and the absorption coefficients $\alpha_{x,y}$. Determination of the total scattered intensity requires an integration over the whole z coordinate. However, many more optical

processes, including the transmission at the sample interface (including the air/sample and sample/substrate surface), absorption and reabsorption of the incident and scattered light in anisotropic crystal system, should be taken into account and makes the situation even more complicated to deal with. On the other hand, one can investigate the thickness-dependent QI effect based on the *ab initio* density functional calculations in collaboration with CID experimental measurement, which should be challenging but worth of a further systematic research effort.

Last but not least, we consider that the quantum interference is generic and ubiquitous an effect in Raman scattering, which is therefore not limited in low-symmetry crystals (*Nature*, 2011, 471, 617-620; *Nano Lett.* 2017, 17, 2381-2388). For example, in the high in-plane symmetry materials such as graphene or TMDC hexagonal monolayers, quantum interference effect should also exist. But these high-symmetry materials would not show chiral response, as we speculate, not only because of no birefringence effect in the materials, but also because of the equivalent quantum interference for LCP and RCP occurring respectively in the K and $-K$ valleys which have the time-reversal symmetry.

In order to address this point in the main text, we updated the following sentences in the last part of Discussion in the revised manuscript:

“Our findings reveal that quantum interference can lead to pronounced chiral response of Raman scattering in materials and indicate that quantum interference can be a generic effect in inelastic optical scattering, which becomes evident when either constructive or destructive interference between all the inelastic scattering pathways dominates, in the condition that the excitation photon energy is larger than the band gap of the materials. This effect is also applicable to the bulk triclinic crystals (ReS₂ and ReSe₂), but additional anisotropic optical environment should be taken into consideration for a complete analysis of chiral response.”

Question 2

They write in the new version of the manuscript that “A quarter wave plate (QWP) is used to produce right-handed (RCP) or left-handed (LCP) circular polarization for excitation. The scattered light passes through the same QWP, and is collected without any analyzer.” I believe this experimental configuration assumes that the Raman scattered light has the same polarization as the

excitation light. However, is that not problematic given that the Raman tensor has off-diagonal elements? I would expect the scattered light to lose its circular polarized nature. They can test this by using an analyzer to see if their Raman scattered light is linearly polarized (parallel to the incident laser) after the quarter-wave plate as they assumed in the current version.

For instance, it is reported in *Nano Lett.* 2015, 15, 5667–5672 that the Raman scattered light from monolayer ReS₂ has both parallel and perpendicular components with respect to the linearly polarized incident 532 nm laser.

Our response:

Thanks for your kind advice. In the new version of the manuscript, the optical setup was placed in Fig. 1a, as show below.

Figure 1. (a) Optical setup of the chiral Raman scattering measurements

Usually in the commercial Raman systems, the polarization states of the incident light cannot entirely retain after passing through the beam splitter or dichroic mirror, especially for the circular polarization. So if we put the QWP before the beam splitter, the circular polarization of the excitation laser would be deteriorated after the light was reflected. This is why we put the QWP directly before the objective lens to produce LCP and RCP.

It can be seen from the optical setup that both the incident and the scattered light would pass through the QWP, but the scattered light was collected without any analyzer, which we emphasized in the manuscript. Although the polarization states were changed by the QWP, but all the photons were collected, and in the manuscript, only the intensities excited by RCP and LCP were compared, but not the polarization states. In other words, we only consider the intensities, but not the polarization states, of the scattered light.

We would like to mention that the polarization of Raman scattered light for the circularly polarized excitation also contains important information on the structure and physical property of materials, and this is another interesting topic. Helicity-resolved Raman spectroscopy (HRRS) has been developed to study the circular polarization of Raman scattering by some specific phonon modes of 2D materials with high symmetry, *e. g.* hexagonal crystal (*Nano Lett*, 2015, 15: 2526–2532; *2D Mater*, 2017, 4: 045002; *PRB*, 2021, 103, 035405). In our group, we also study the HRRS of a series of 2D materials including graphene and MoS₂. In the case of ReS₂, the scattered light loses its circular polarized nature, as the referee pointed out. This can be explained by the equations of the Raman efficiency of ReS₂ according to the Raman selection rules as below.

For the co-polarized Raman spectra, the Raman efficiency is represented as

$$I_{RR} = I_{LL} \propto a^2 + c^2 + 2ac \cdot \cos(\varphi_a - \varphi_c)$$

However, in the cross-polarized Raman spectra, there are two distinct conditions. With the LCP excitation and RCP analyzation, the Raman scattering efficiency is

$$I_{LR} = a^2 + 4b^2 + c^2 + 4ab\sin(\varphi_a - \varphi_b) + 4bc\sin(\varphi_b - \varphi_c) - 2accos(\varphi_a + \varphi_c)$$

When the excitation polarization is RCP, and LCP analyzation, the efficiency is

$$I_{RL} = a^2 + 4b^2 + c^2 - 4ab\sin(\varphi_a - \varphi_b) - 4bc\sin(\varphi_b - \varphi_c) - 2accos(\varphi_a + \varphi_c)$$

In conclusion, with circularly polarized excitation, the polarization states of the scattered light have both co-polarized and cross-polarized components.

Reviewers' Comments:

Reviewer #1:

Remarks to the Author:

I thank the authors for clarifying the experimental setup and the distinction between the 1L and bulk.